# Convective/Stratiform Precipitation Classification Using Ground-Based Doppler Radar Data Based on the K-Nearest Neighbor Algorithm

**Zhida Yang, Peng Liu and Yi Yang *** 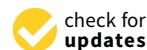

Research and Development Center of Earth System Model (RDCM), College of Atmospheric Sciences, Lanzhou University, Lanzhou 730000, China; yangzhd17@lzu.edu.cn (Z.Y.); liup16@lzu.edu.cn (P.L.)

* Correspondence: yangyi@lzu.edu.cn

**Abstract:** Stratiform and convective rain types are associated with different cloud physical processes, vertical structures, thermodynamic influences and precipitation types. Distinguishing convective and stratiform systems is beneficial to meteorology research and weather forecasting. However, there is no clear boundary between stratiform and convective precipitation. In this study, a machine learning algorithm, K-nearest neighbor (KNN), is used to classify precipitation types. Six Doppler radar (WSR-98D/SA) data sets from Jiangsu, Guangzhou and Anhui Provinces in China were used as training and classification samples, and the 2A23 product of the Tropical Precipitation Measurement Mission (TRMM) was used to obtain the training labels and evaluate the classification performance. Classifying precipitation types using KNN requires three steps. First, features are selected from the radar data by comparing the range of each variable for different precipitation types. Second, the same unclassified samples are classified with different k values to choose the best-performing k. Finally, the unclassified samples are put into the KNN algorithm with the best k to classify precipitation types, and the classification performance is evaluated. Three types of cases, squall line, embedded convective and stratiform cases, are classified by KNN. The KNN method can accurately classify the location and area of stratiform and convective systems. For stratiform classifications, KNN has a 95% probability of detection, 8% false alarm rate, and 87% cumulative success index; for convective classifications, KNN yields a 78% probability of detection, a 13% false alarm rate, and a 69% cumulative success index. These results imply that KNN can correctly classify almost all stratiform precipitation and most convective precipitation types. This result suggests that KNN has great potential in classifying precipitation types.

**Keywords:** precipitation classification; K-nearest neighbor; Doppler radar; Tropical Precipitation Measurement Mission (TRMM)

## 1. Introduction

Precipitation can be divided into stratiform precipitation and convective precipitation [1]. A convective precipitation system generally has the characteristics of strong upward motion, small areal coverage and high precipitation intensity, while a stratiform precipitation system has the characteristics of weak upward motion, large areal coverage and weak precipitation intensity. The classification of precipitation can be used in meteorological research, weather forecasting and meteorological disaster prevention. First, there are different precipitation growth mechanisms and different physical principles between convective and stratiform precipitation. Research on convective and stratiform classification can provide a better understanding of the physical mechanisms of clouds. Additionally, convective systems have an important influence on the thermal balance of the atmosphere [2,3],

and thermodynamic differences can lead to different latent heat distributions, moisture cycling, and cold rain and warm rain processes, which can have different effects on cloud lifetime and Earth's climate [1,4]. The energy of a convective system is often expressed in terms of the apparent heat sources and apparent moisture sinks. An apparent heat source is difficult to measure but can be estimated by precipitation [5,6], and different types of precipitation can reflect different thermodynamic structures. Finally, precipitation estimation plays an important role in understanding the hydrological cycle reducing uncertainties in global climate change model predictions for future environmental scenarios, weather forecasting and disaster prevention [7–9]. In the traditional method of precipitation estimation, a Z-R relationship is adopted, and the estimated precipitation is calculated through the relation between the radar echo intensity and precipitation. However, different types of clouds have different structures and precipitation growth mechanisms, and the use of a single Z-R relation cannot provide a good estimate of precipitation. It is helpful to improve the accuracy of precipitation estimation by classifying precipitation clouds and adopting different Z-R relations for different types of clouds [10,11]. However, there is no clear boundary between stratiform precipitation and convective precipitation, and it is difficult to distinguish one from the other directly from radar data. As a result, a number of methods have been developed to classify precipitation types.

In an early study, ground rain gauges were used for classification. This method classifies rainfall as convective when the gauge data exceed some background level by a certain amount [12,13]. The background-exceedance technique (BET) uses radar reflectivity to identify the convective core in a certain plane and set the radius of influence; the area inside the influence radius is considered the convective rain zone, and the area outside the influence radius is the stratiform rain zone [14]. Steiner, et al. [15] modified the BET method using a variable radius of influence and a variable threshold instead of a fixed radius of influence and threshold in the BET method. The authors proposed the method in 1995, and the method was named SHY95 using the initials of the three authors and the year that the method was proposed. An extended SHY95 method was applied by DeMott, et al. [16], who used a two-dimensional BET at each height level within a volume of radar reflectivity to extend this approach to three dimensions. They suggested that using low-level data may lead to the misclassification of convective cells that tilted strongly with height and showed that using three-dimensional data can improve the accuracy of precipitation classification. Biggerstaff and Listemaa [5] modified the classification results of SHY95 by considering the vertical structure of the radar reflectivity factor based on the SHY95 method and found that the method yielded higher accuracy than SHY95. Bringi, et al. [17] classified precipitation types by calculating the standard deviation of the drop size distribution (DSD). When the standard deviation of the DSD is smaller than a certain standard, it is classified as stratiform precipitation, and when the standard deviation of the DSD is larger than this standard, it is classified as convective precipitation. Instead of using the traditional method based on the BET, Anagnostou [18] proposed an algorithm for classifying stratiform and convective clouds using an artificial neural network (ANN). The cloud-top height, reflectivity at a height of 2 km and 4 other features were used in the ANN training. Compared with other traditional algorithms based on the BET, the ANN exhibited better performance. The DSD has also been used to classify precipitation [19]. Based on a large number of rain events and by computing the Z-R relationship, the average DSD and the corresponding parameters, microphysical analysis can be performed; the rain distribution and precipitation type can be adequately characterized by a gamma DSD. Zhang and Qi [20] developed a method that automatically corrects for large errors due to the bright band in a real-time national radar quantitative precipitation estimation product, and the performances were good [21–23]. Yang, et al. [24] applied the fuzzy logic (FL) method for precipitation classification research using the 2-km height echo reflectivity, vertical integral liquid water content and other characteristics for classification, and the FL classification results were more natural and realistic than those of other methods. Yang, et al. [25] used FL to classify precipitation types and estimate precipitation. The results showed that compared with the Z-R relationship, FL can reduce the underestimation of precipitation and improve the accuracy of estimating precipitation using radar data.

Some studies have used satellite data to classify precipitation types. Adler and Negri [26] used infrared satellite data and applied a variant of the BET to classify convective and stratiform precipitation. Unlike convective cores denoted by radar reflectivity in the BET, they used the minimum cloud-top temperatures to identify the convective core area. The radius of influence of each core was dependent of the magnitude of the infrared brightness temperature of the core [26,27]. Goldenberg, et al. [28] used an infrared cloud-top temperature method similar to the BET to classify convective and stratiform precipitation for a tropical cloud cluster. Awaka, et al. [29] used TRMM precipitation radar data. Two algorithms, the vertical contour mode (V-method) and horizontal contour mode (H-method), were used in the study to classify precipitation types. If the classification results of the two algorithms are the same, the classification result is determined, and if the classification results differ, fusion-based classification results are used. The V-method can be used to detect the bright band. Once the bright band is detected, the precipitation type is classified as stratiform precipitation. Then, the V-method continues to detect convective precipitation according to the radar reflectivity. If the precipitation type is neither stratiform nor convective, it is classified as another type of precipitation. The H-method is based on Houze's classification model [15] using the horizontal echo intensity at a height of 2 km to assess the type of precipitation.

The precipitation process involves complex thermodynamic mechanisms and cloud microphysical mechanisms during sedimentation. These principles are difficult to fully explore. Thus, it is difficult to classify precipitation types based on these mechanisms. Machine learning can be used to build models and capture the characteristics of data such that changes in the data can be predicted and the data can be classified into different categories based on the relevant characteristics. When using machine learning to classify precipitation types, it is not necessary to understand the precipitation mechanisms of convective precipitation or stratiform precipitation, and the representative and appropriate variables for classification and labeling can be selected to achieve optimal classification. Machine learning is a discipline that uses experience to improve the performance of a system by means of calculations [30]. In computer systems, experience often exists in the form of data; thus, the main area of machine learning research involves computer algorithms that generate models from data. The main types of machine learning include supervised learning, unsupervised learning and semisupervised learning. Supervised learning is a method of adjusting the parameters of a model with a set of known classes of samples to achieve the required performance. Supervised learning includes decision trees, boosting and bagging algorithms, support vector machines, etc. Semisupervised learning refers to the fact that data sets contain both identified and unidentified data, and unidentified data are obtained using the identified data. Semisupervised learning usually includes semisupervised Support Vector Machine (SVM), semisupervised clustering, etc. In unsupervised learning, training samples do not have known characteristic information. Unsupervised learning reflects the inherent nature and laws of data by learning unlabeled training samples, providing a basis for further data analysis. This approach is commonly used in clustering.

The K-nearest neighbor (KNN) method is a type of supervised learning algorithm that has been widely used in pattern recognition and classification. KNN relies on the nearest k samples instead of all the samples for classification and is most suitable for classifying samples with overlap and unclear boundaries. KNN was proposed by Fix and Hodges [31], and Cover and Hart [32] further developed and improved the algorithm. KNN has fewer tunable parameters and provides faster calculations for small data sets than other methods. Thus, this approach has advantages in solving classification problems involving precipitation types. Machine learning is seldom used to classify precipitation types. KNN is a mature classification algorithm with many advantages and has been used in many fields, but there is no relevant study to prove the applicability of KNN in the classification of precipitation types, and the present study attempts to use and explore the applicability of KNN to classify precipitation types.

This paper consists of the following: Section 2 introduces radar data and satellite data used in this paper, Section 3 describes the implementation process of the KNN algorithm and the performance of

the selected variables under different conditions, Section 4 presents the results of the KNN classification of precipitation type, and the final section provides a summary and conclusion.

## 2. Data Description

The Doppler radar data used in this study are from the six S-band China Next-Generation Weather Radars (CINRAD/SA), and the site information and usage period of the radars are shown in Table 1. The radars are 10-cm wavelength Doppler radars with a 1° half-power beam width. The radar data consist of volume scans of the radar reflectivity, average radial velocity and spectral width. The radars are operated in 360° azimuthal volume scan mode with steps in elevation angles from 0.5° to 19.5° during periods of precipitation. The number of elevation steps and temporal resolution of the data depend on the operational mode of each radar. The radial bin spacing is 250 m. The radar data used in this paper are interpolated by the Barnes interpolation algorithm to a horizontal grid with a resolution of 1 km × 1 km [33] and a vertical resolution of 500 m over a depth of 18 km in the Cartesian coordinate system. The origin of the coordinates is the position of the radar. The data are quality controlled.

The precipitation radar (PR) is mounted on the TRMM satellite. The system takes 92.5 min to scan Earth, and it can scan Earth 16 times a day. The scanning range is from 38°N to 38°S and 180°W to 180°E, and the scanning swath width is 247 km. The spatial resolution is 5 km. As TRMM uses a low-altitude orbit, the PR can provide measurements of 3D rainfall distributions with unprecedented accuracy in the tropics and subtropics. The products of TRMM have been widely used in a variety of studies, such as the study of precipitation distribution patterns in tropical and subtropical regions [34], to improve the accuracy of precipitation prediction [35]. Research on precipitation structure and properties [36] has demonstrated the reliability of TRMM and its products. In addition to the basic information provided by the PR, the 2A23 product includes rain characteristics observed by the PR. Based on the high vertical resolution of the PR data, the 2A23 product can accurately detect the bright band (BB) occurrence and its height. The following variables are used in this paper: rain flag, which indicates the possibility of precipitation in a grid, the rain type, which is the classification of the precipitation type, including stratiform, convective and others, and the height of the bright band, which indicates whether a BB is detected in a grid and the height of the BB if there is one. As warm and cold rain precipitation are not directly classified and interpreted in the 2A23 product, the classification results do not include the classification of cold or warm rain. The precipitation radar has a wavelength of 2.2 cm, and the ground-based radar used in this paper has a wavelength of 10 cm. Therefore, the precipitation radar will be subject to more two-way path attenuation. In addition, the scanning angle, signal frequency and sensitivity of ground-based radar differ from the PR. The main purpose of this paper is to classify the types of precipitation, taking the 2A23 precipitation classification product of the PR instead of the echo reflectivity data of the PR as the training sample label for KNN and evaluating the training results; these differences are not taken into consideration and have no effect on the results.

The 2A23 product has a horizontal resolution of 5 km, and the horizontal resolution of the radar data is 1 km. To make these two datasets comparable, the interpolation scheme and data selection are described below.

Instantaneous 2A23 data and ground radar data that are within a time lag with a maximum of 3 min are projected into a Cartesian coordinate with 5 km × 5 km horizontal resolution. Each ray of a PR swath is projected on the Cartesian grid by the status of the nearest pixel.

There are still steps needed to make the comparisons of two datasets meaningful. These steps are as follows: (1) a pixel is classified as stratiform by the 2A23 product if a BB is not detected and ref2km is greater than 40 dBz or if there is a BB detected and ref2km is greater than 42 dBz with a horizontal gradient greater than 3 dB/km; (2) a pixel is classified as convective by the 2A23 product if no BB is detected but ref2km is less than 40 dBz; and (3) a pixel is classified as convective by the 2A23 product if a BB is detected.

**Table 1.** Radar site information and data usage time.

| Station | Date | Coordinate | Usage | Cases Number |
|---|---|---|---|---|
| Hefei | 6 June 2010–10 June 2010 | 117.258°E, 31.867°N | Classification | 2 |
| Fuyang | 25 June 2005–26 June 2005. 7 July 2007–9 July 2007 | 115.741°E, 32.879°N | Training and Classification | 4 |
| Lianyungang | 1 July 2012–31 July 2012 | 119.294°E, 34.651°N | Training and Classification | 7 |
| Nanjing | 1 July 2012–31 July 2012 | 118.698°E, 32.191°N | Classification | 5 |
| Guangzhou | 4 June 2008–13 June 2008 | 120.976°E, 32.076°N | Training and Classification | 4 |
| Wenzhou | 4 June 2008–13 June 2008 | 117.152°E, 34.293°N | Classification | 2 |

## 3. Algorithm and Features

### 3.1. Overview of the K-Nearest Neighbor Method

KNN is a classification algorithm used to classify precipitation types in this paper. KNN does not have a display learning process. In the training phase, KNN simply saves the training samples and processes them after receiving the test samples [37]. Input samples with classification labels are used as KNN inputs for training samples. To achieve satisfactory classification results, a larger number of training samples are needed, and the proportion of each classification in the training samples should be as uniform as possible. In the actual precipitation process, the spatial and temporal extents of stratiform precipitation are usually larger than those of convective precipitation. In the interpolated and screened samples, the number of stratiform precipitation grid points is much larger than the number of convective precipitation grid points. If such data sets are used as KNN training samples, the classification results will be generally biased toward stratiform precipitation. The number of different types of precipitation samples in the training sample needs to be adjusted. Samples of different types of precipitation were randomly selected, and the training sample set was reconstructed according to stratiform cloud precipitation, convective precipitation and other precipitation with a ratio of 1:1:1.

When there are samples to be classified, to obtain the classification results, the distance between the sample to be classified and all the training samples is calculated. After calculating the distance, k training samples with the smallest distances from the sample to be classified are selected. The k training samples have the same influence factor, and the probability that the sample is classified as type j is as follows:

$$P_j = \frac{N_j}{k}, \tag{1}$$

where $P_j$ is the probability that the sample is classified as type $j$ and $N_j$ is the number of training samples with a classification label of type $j$ among the k-nearest training samples. When the $P_j$ value is a maximum, type $j$ is the classification result.

### 3.2. Selection of Features

Using KNN to classify different precipitation types requires that the variables used in the classification have significant differences for different precipitation types, such as stratiform and convective precipitation, so that the precipitation types can be well distinguished. Using the 2A23 product as a reference, the variations in the frequency of the variables used for the classification of different precipitation types is determined and compared horizontally to validate the classification variable. However, if the bright band is present, the reflectivity will increase significantly, which will negatively influence the classification results. The bright band is not expected to appear at the time of classification. An altitude of 2 km is high enough to provide a sufficient amount of radar data out to a

radius of approximately 150 km, and a 2-km altitude is low enough to avoid serious effects of the bright band, which usually appears at a height of 2.5 km to 4.5 km in tropical and sub-tropical areas [38].

Feature 1: Horizontal distribution characteristics of radar reflectivity at a height of 2 km (ref2km) [18]. ref2 km can often reflect the horizontal structural characteristics of convective systems. For stratiform systems, this height should be adjusted appropriately. In some cases, the temperature at 2 km in the vertical height layer is close to 0 °C, and there is a mixture of liquid and solid phase water and transitions between the two phases. During the conversion process of solid water to liquid water, a water-coating film is formed on the surface of melting water, and the difference between the negative refractive index values of the liquid phase particles and the solid phase particles will cause the back reflectance measured by the radar to increase, which may result in a flat and strong echo band. If such strong echo bands are not distinguished, it may cause an erroneous assessment of that type of precipitation in the area. Figure 1a is the frequency distribution diagram of ref2km. The frequencies of stratiform precipitation and convective precipitation increase below 30 dBz. In the range of 30–35 dBz, the frequency of stratiform precipitation reaches a maximum, and then the frequency decreases gradually with larger reflectivity values. At 40–45 dBz, the frequency drops to almost zero. The convective precipitation reaches a maximum frequency at 35–40 dBz, and the frequency decreases gradually with larger reflectivity values. However, when this frequency is above 50 dBz, there is still convective precipitation. The frequency graph of ref2km shows that the value of ref2km exhibits large differences and can be used to sufficiently discriminate among different precipitation types. It is reasonable to use ref2 km for the classification of precipitation types.

Feature 2: Vertically integrated liquid-water content (VIL) [39]. The liquid water content M and radar reflectivity Z can be defined as follows:

$$\mathrm{M} = \frac{\rho_w \pi}{6} \int_0^x n(a)a^3 da, \tag{2}$$

$$Z = \int_0^x n(a)a^6 da, \tag{3}$$

where $x$ is the maximum drop diameter, and $\rho_w$ is the density of water. When the Marshall-Palmer drop size distribution is used in Equations (3) and (4), the error is small if the upper limit of integration, $x$, is replaced by $\infty$.

$$\mathrm{M} = \frac{N_0 \rho_w \pi}{6} \int_0^\infty \exp(-ba)a^3 da = \frac{N_0 \rho_w \pi}{6} \frac{\Gamma(4)}{b^4} = \frac{N_0 \rho_w \pi}{b^4}, \tag{4}$$

$$Z = N_0 \int_0^\infty \exp(-ba)a^3 da = \frac{N_0 \Gamma(4)}{b^7} = \frac{720 N_0}{b^7}. \tag{5}$$

Eliminating the parameter b in Equations (5) and (6) yields

$$\mathrm{M} = \frac{N_0 \rho_w \pi}{[720 \times 10^{18} N_0]^{4/7}} Z^{4/7}. \tag{6}$$

For $N_0 = 8 \times 10^6 \ \mathrm{m}^{-4}$ and $\rho_w = 10^6 \ \mathrm{g/m}^3$,

$$\mathrm{M} = 3.44 \times 10^{-3} Z^{4/7}, \tag{7}$$

where the units of M are $\mathrm{g/m}^3$ and those of Z are $\mathrm{mm}^6/\mathrm{m}^3$.

$$M^* = \int_{h_{base}}^{h_{top}} M dh' = 3.44 \times 10^{-6} \int_{h_{base}}^{h_{top}} Z^{\frac{4}{7}} dh', \tag{8}$$

　　　Here, $M^*$ is VIL, which is given in units of kg/m$^2$; Z is radar reflectivity, with units of mm$^6$/m$^3$; and $h_{top}$ and $h_{base}$ are the uppermost and lowermost layers of the radar echo, with units of meters. VIL reflects the overall vertical state of the echo area, and it is possible to filter the effects of false high echoes caused by bright bands and topographical factors. At the same time, changes in VIL are a good reflection of changes in a convective system. However, in nonconvective areas, VIL changes little, and the reference value decreases accordingly. Figure 1b shows the frequency distribution of VIL. The frequency of stratiform precipitation reaches a maximum for VIL of 2 kg/m$^2$ and then decreases rapidly. The VIL value of conditions with almost no stratiform precipitation could reach 4 kg/m$^2$. In contrast, the frequency of convective precipitation reaches a maximum near a VIL value of 4 kg/m$^2$ and then decreases, although convective precipitation exists even if the VIL value reaches 18 kg/m$^2$. Additionally, VIL considerably varies and sufficiently reflects different precipitation types. Thus, it is reasonable to use VIL for the classification of precipitation types.

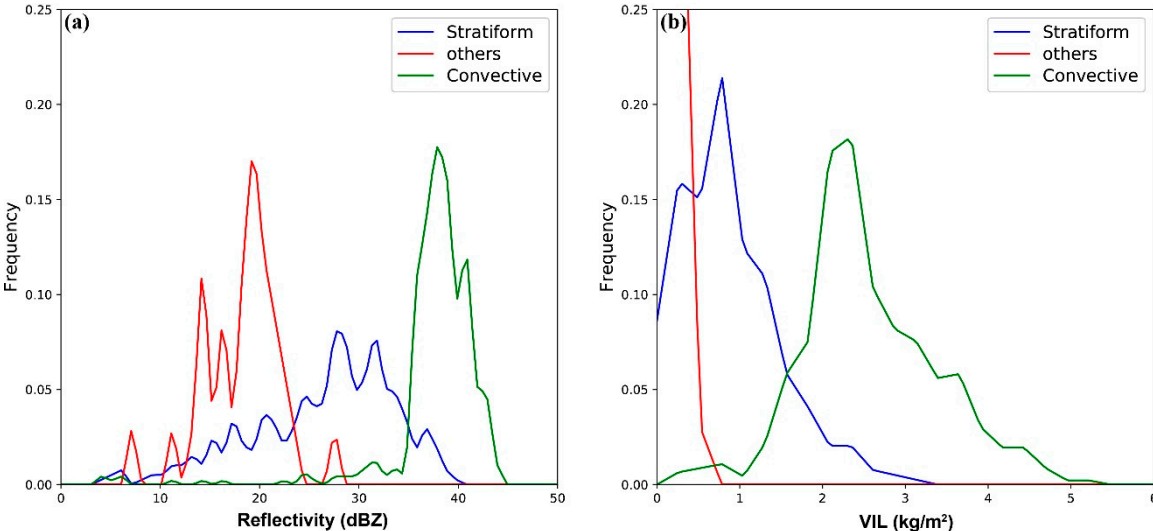

**Figure 1.** The variations in different types of precipitation frequency as a function of (**a**) Horizontal distribution characteristics of radar reflectivity at a height of 2 km (ref2km) and (**b**) Vertically integrated liquid water content (VIL).

　　　Variables ref2km and VIL have different scales. During precipitation, ref2 km usually has a minimum of 16 dBz and maximum of 50 dBz, while VIL has a minimum of 0 kg/m$^2$ and maximum of 10 kg/m$^2$. When calculating the Euclidean distance, the effect of VIL on the distance can be significantly small due to the smaller scale. Thus, the data need to be normalized or standardized before calculating the distance, which could decease the influence of variables with different scales.

　　　The standardized Euclidean distance can decrease the influence of variables with different scales by standardizing the data. The standardized Euclidean distance between sample x and sample y is calculated as follows:

$$d(x, y) = \sqrt{\sum_{i=1}^{n} \frac{(x_i - y_i)^2}{s_i^2}}, \tag{9}$$

where $s_i$ is the standard deviation of $x_i$ and $y_i$ over the sample set.

　　　The Euclidean distance, Manhattan distance, and standardized Euclidean distance are used to classify cases at the same time. Although the scales of ref2 km and VIL are not the same, the classification results of the standardized Euclidean distance, Euclidean distance and Manhattan distance do not differ substantially. To remove the possible occurrence of unstandardized adverse effects, the standardized Euclidean distance is used as the distance in the KNN in this study.

### 3.3. Training and Classification

In this paper, two variables, ref2km and VIL, are used as classification variables, and the corresponding precipitation classification results from the 2A23 product, as classification labels, are put into the KNN algorithm training process. An appropriate k value is selected, and ref2km and VIL of the sample to be classified are put into the KNN. The standardized Euclidean distance between the sample to be classified and each training sample stored in the KNN is calculated. Then, the class with the largest number of k training samples is taken as the classification result.

## 4. Results

In this section, typical independent individual cases are selected to determine whether the classifications are correct. In addition, an overall analysis is used to assess the performance of the KNN algorithm.

### 4.1. Evaluation Method

The KNN classification results were compared with the 2A23 product, and the results were evaluated based on the probability of detection (POD), false alarm rate (FAR), and cumulative success index (CSI):

$$\text{POD} = \frac{n_s}{n_s + n_f}, \tag{10}$$

$$\text{FAR} = \frac{n_{fa}}{n_s + n_{fa}}, \tag{11}$$

$$CSI = \frac{n_s}{n_s + n_f + n_{fa}}, \tag{12}$$

In the above three formulas, $n_s$, $n_f$ and $n_{fa}$ are the numbers of successful classifications, failed classifications and false alarm classifications, respectively. Success is counted when a method classification is similar to the PR 2A23 classification, failure is counted when a classification is not similar to the PR 2A23 class, and false alarm is counted when a pixel is classified opposite the PR 2A23 classification.

The POD can reflect the relationship between the number of successful classification points and the number of failed classification points; the higher the POD value is, the better the classification performance. The FAR can explain the proportion of false alarm points in the classification according to the number of correct points in the classification results. The lower the FAR value is, the better the classification performance. The CSI reflects the overall classification performance; it can explain the proportion of correctly classified points among all classified points, and when the CSI reaches a high value, the classification performance is satisfactory.

### 4.2. K Value

For a finite set classification, the classification error rate of the KNN tends to converge to a certain value as k increases [40]. When k is too large, the classification accuracy rate does not increase significantly, which results in wasted computational resources. When k is too small, the classification accuracy rate is low. Choosing the right k value helps improve the classification accuracy and reduce the calculation amount to improve the calculation speed.

Figure 2 shows the classification of an embedded convective process in the Guangzhou area at 05:28 (UTC) on 6 June, 2008, using the standardized Euclidean distance as the calculated distance. The effect of using different k values on the overall classification results is small. At the junction of different types of precipitation, the results of different k classifications are slightly different. When k is equal to 5, the boundary between stratiform and convective precipitation is rough, and when k is chosen to be 10 or more, the boundary is smooth.

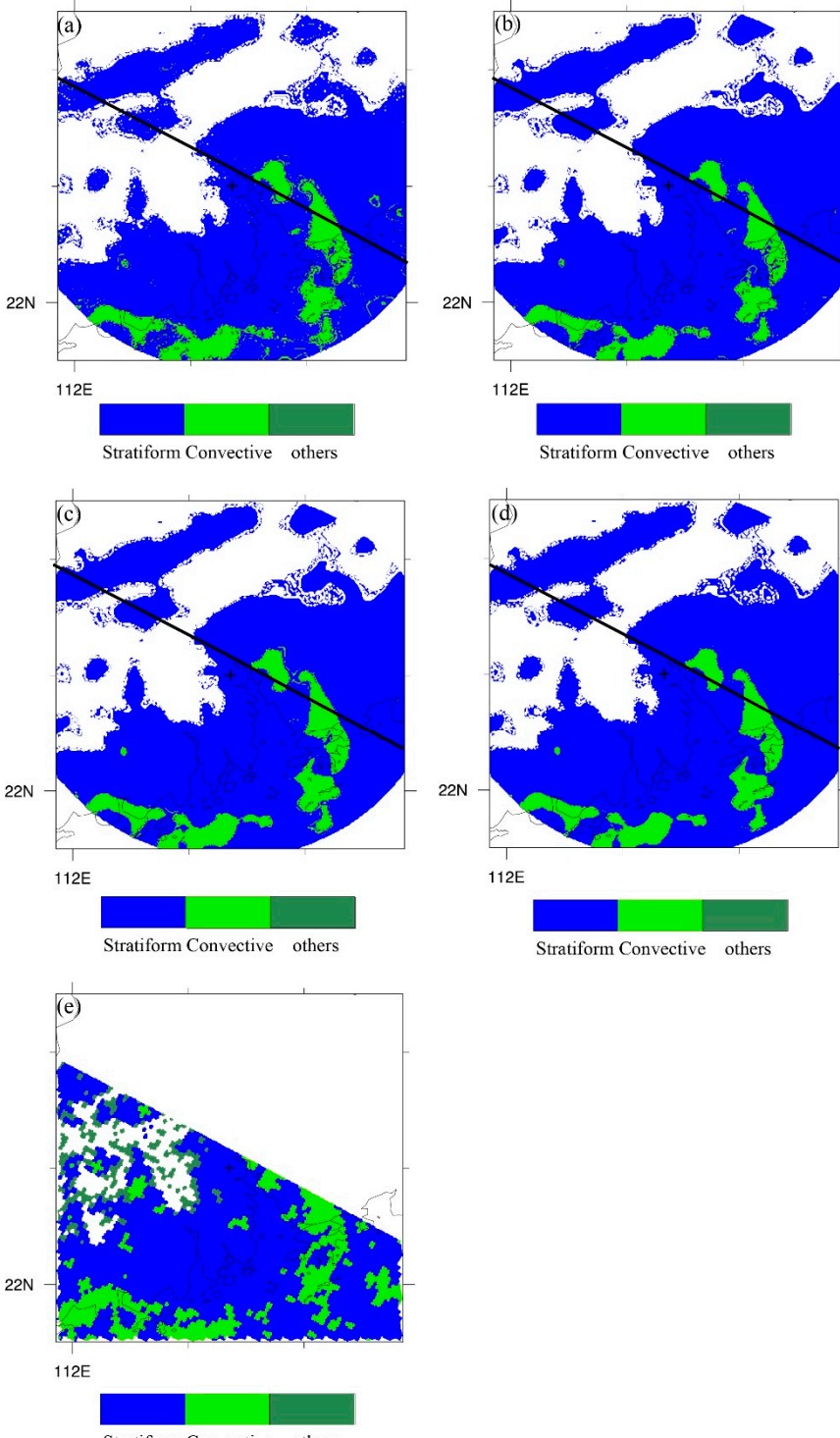

**Figure 2.** A case from Guangzhou at 05:28 (UTC) on 6 June, 2008, shows the classification results for different k values. (**a**–**d**) are the results for k = 5, 10, 15, and 20, and (**e**) is the classification of the 2A23 product. The bold black line represents the boundary of the PR scan range.

Other cases from Anhui, Jiangsu and Wenzhou were selected for analysis. The results are shown in Table 2. Although the classification result boundary is rough when k is equal to 5, this value yields the highest POD and CSI and a low FAR among several different k values. When k is equal to 10, the smallest FAR is observed, although the POD is not high and the CSI is low. When k is greater than

10, the POD, FAR and CSI differ, although the difference is not obvious. Thus, when k is equal to 5, the performance is obviously the best, therefore, the value of k is set to 5 in this paper.

**Table 2.** The POD, FAR and CSI for different k values in the same case.

| K | POD | FAR | CSI |
|---|-----|-----|-----|
| 5 | 0.461 | 0.269 | 0.394 |
| 10 | 0.385 | 0.228 | 0.346 |
| 15 | 0.409 | 0.244 | 0.362 |
| 20 | 0.390 | 0.234 | 0.349 |

The classification of precipitation types for different types of weather processes can fully reflect the KNN classification performance. Squall line cases, embedded convective cases and stratiform cases are selected for KNN classification analyses.

*4.3. Squall Line Case*

Figure 3 shows a squall line case. Figure 3a shows the 2A23 product. Two northeast-southwest-oriented convective belts are classified within the scanning range. There are tiny gaps between the two band-shaped convective cells. Two northeast-southwest-oriented convective belts are classified within the scanning range. The cluster of convective cells is independent of the band-shaped cells. Outside the convective cells, stratiform precipitation covers large area. The southeastern part of Figure 3a is classified as an unknown type of precipitation. In this case, precipitation may occur, although the type of precipitation is unknown. Figure 3b shows the results of the KNN classification. There is a band-shaped northeast-southwest-trending convective cell, which is observed in the 2A23 product. However, the boundary between the two band-shaped convective cells is not obvious. In the northeast direction of the band-shaped convective cell, a cluster of convective cells is also classified, and the cluster shape is similar to that of the 2A23 product. There is also a massive convective cell in the northeast portion of the cluster of convective cells. In the 2A23 product, due to the sweep coverage, there are no corresponding data for this area. The northeast corner of the radar corresponds to the area classified as unknown in the 2A23 product. Because the KNN categorical variable data have no values in that area, no classification is provided. The southwest corner of Figure 3b is a void area due to the radar elevation angle.

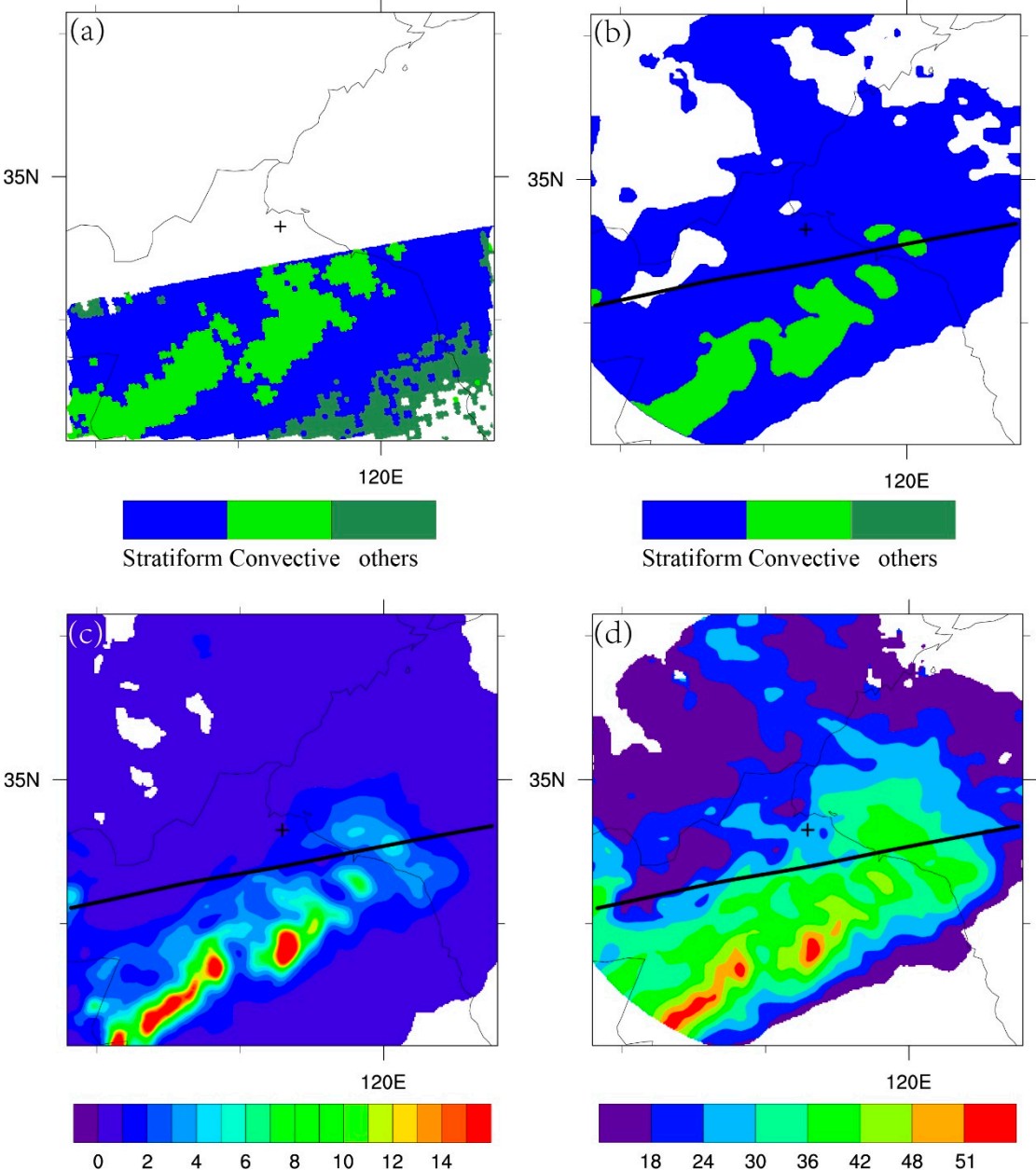

**Figure 3.** A squall line in Lianyungang at 13:25 (UTC) on July 4, 2012: (**a**) is the classification of the 2A23 product, (**b**) is the KNN classification, (**c**) is VIL, and (**d**) is ref2km. The bold black line represents the boundary of the PR scan range.

Figure 3c shows the VIL. In Figure 3c, there is a northeast-southwest band-shaped high-value area. There are multiple independent high-value centers in the high-value area. The values of all these centers exceed 14 kg/m$^2$. The value near the center also reaches or exceeds 4 kg/m$^2$, and there is a block-shaped high-value area in the northeast of the band-shaped high-value area, the value of which exceeds 6 kg/m$^2$. Additionally, in the northeastern part of the high-value area, there is an area with values exceeding 4 kg/m$^2$. The VIL values in the other areas are less than 2 kg/m$^2$. Figure 3d shows ref2km. The high-value area in the figure corresponds to the high-value area in Figure 3c, and the value of each high-value center exceeds 50 dBz. To the northeast of the band-shaped high-value area, there are also areas exceeding 40 dBz. The two high-value areas in the northeast direction of the band-shaped high-value area in Figure 3c,d are consistent with the area classified as convective by the KNN algorithm.

### 4.4. Embedded Convective Case

Figure 4 shows the classification results for an embedded convective scenario. In Figure 4a, an arched area in the center of the figure is classified as convective by the 2A23 product, and a large area on the west side of the arched convective area is classified as an unknown type of precipitation. There are small stratiform precipitation areas in the northwest corner and a large stratiform precipitation area on the east side of the arched convective cell. Due to the scope of the sweeping surface, there are no data available for the south side. The arched area in the center of Figure 4b is classified as convective precipitation. There is also a convective precipitation area outside the 2A23 product range, and there is a clear boundary between the two arched convective cells. There are large stratiform areas in the northeast portions of the two arched convective cells, and there are stratiform areas in the northwestern parts of the convective cells. The shape and location of the scattered stratiform areas are consistent with those in the 2A23 product. Most of the areas classified as unknown precipitation in the 2A23 product are due to missing values for the variables used for the classification. In Figure 4c, there are two arched high-value areas. The VIL values of the two high-value areas are greater than 4 kg/m$^2$, and there are obvious gaps between the two arched high-value areas. The VIL of the interval area is between 2 kg/m$^2$ and 3 kg/m$^2$. On the northeast side of the high-value area, the VIL is above 2 kg/m$^2$, and in some other areas, the value is more than 3 kg/m$^2$. These areas are classified as stratiform in the 2A23 product. There are scattered blocks with VIL values exceeding 1 kg/m$^2$ on the west side, the northwest side and the south side of the arched area, and the remaining VIL values are all below 0.5 kg/m$^2$. In Figure 4d, the radar reflectivity at the corresponding position of the high-value area in Figure 4c exceeds 36 dBz, and the reflectivity in the northeastern area of the arched high-value area exceeds 24 dBz. The reflectivity in some of this area exceeds 30 dBz, and in the scattered block area near this arched area, the reflectivity also reaches or exceeds 24 dBz.

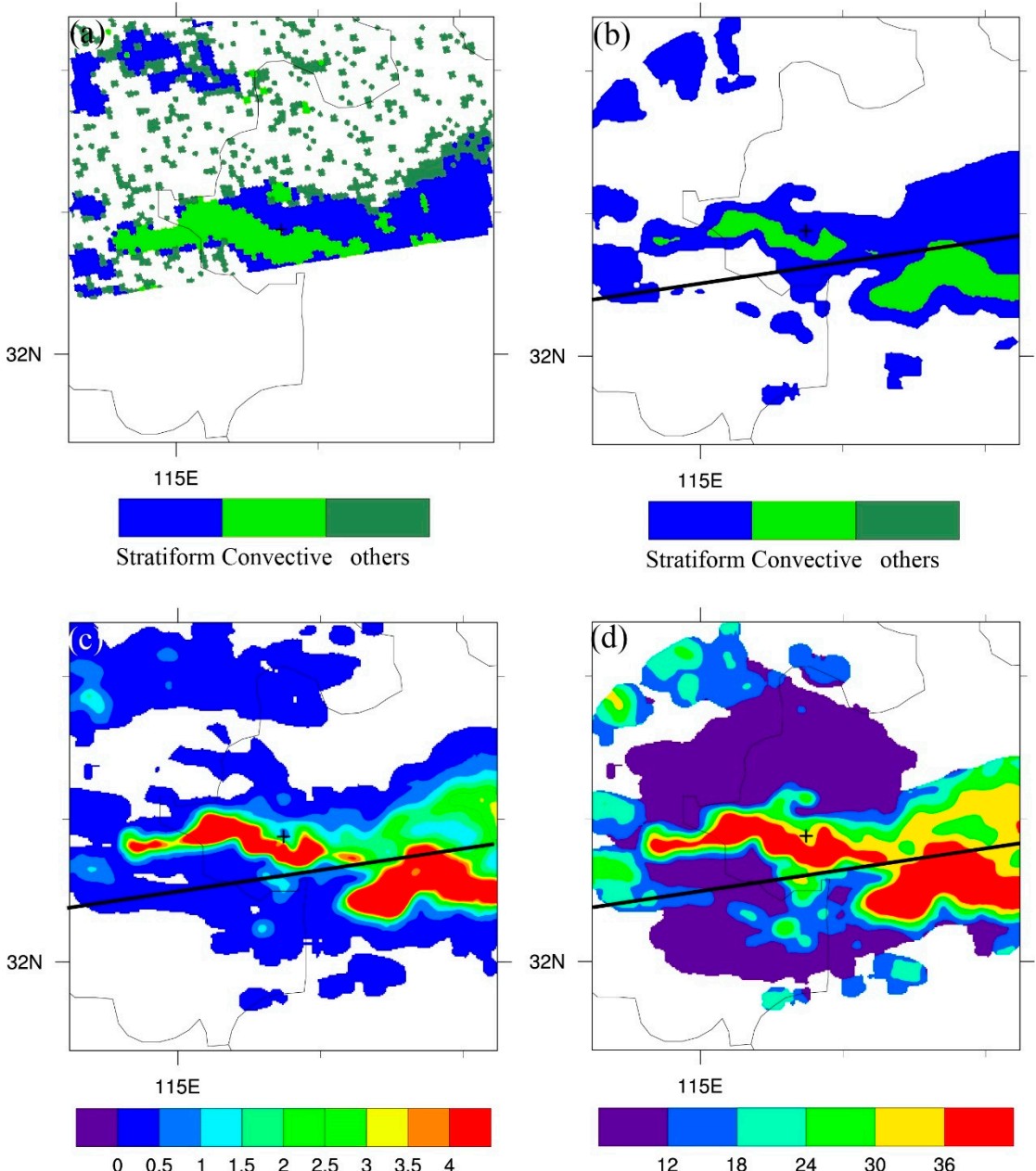

**Figure 4.** An embedded convective system in Fuyang at 01:41 (UTC) on 8 July 2007: (**a**) is the classification of the 2A23 product, (**b**) is the KNN classification, (**c**) is VIL, and (**d**) is ref2km. The bold black line represents the boundary of the PR scan range.

### 4.5. Stratiform Case

Figure 5 shows the classification result of a stratiform case. In Figure 5a, a large northwest-southeast-trending band-shaped area is classified as a stratiform area by the 2A23 product, and a small stratiform block is classified on the northwest side of the band-shaped area. In addition, parts of this area are classified as unknown or no precipitation areas. The southern part of the figure is beyond the PR scanning range; thus, there are no data in this area for the 2A23 product. The north side of the solid black line in Figure 5c is within the PR satellite scanning range, and the area and shape of the region classified as stratiform in this range are consistent with those of the 2A23 product. In Figure 5c, the VIL value of the northwest-southeast-trending area exceeds 0.5 kg/m$^2$, and the VIL value of the high-value area exceeds 1 kg/m$^2$. The VIL values of other areas are less than 0.5 kg/m$^2$. The reflectivity of the areas in Figure 5c,d is greater than 18 dBz, with some areas exceeding 24 dBz.

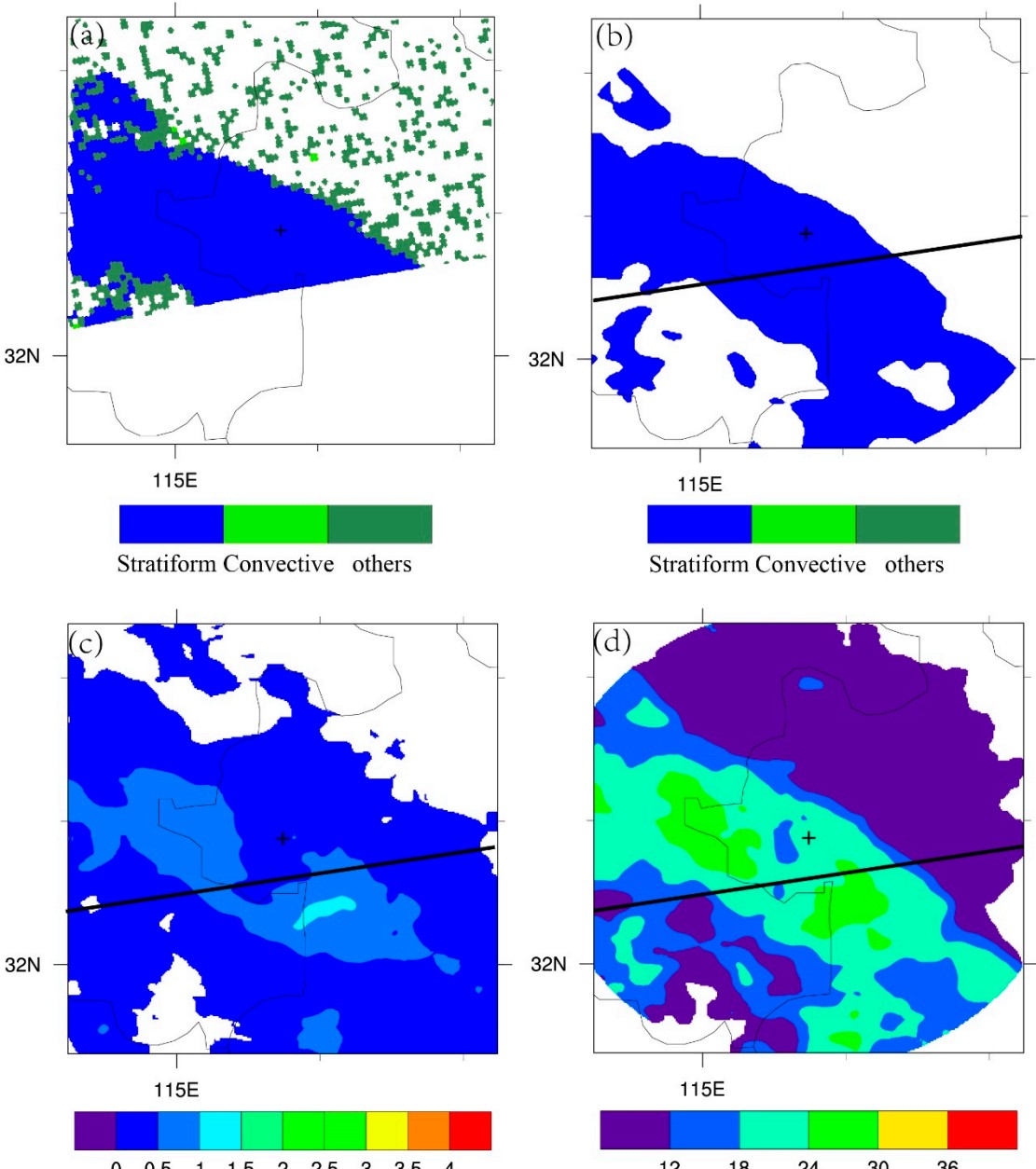

**Figure 5.** A stratiform precipitation system in Fuyang at 13:47 (UTC) on 7 June 2010: (**a**) is the classification of the 2A23 product, (**b**) is the KNN classification, (**c**) is VIL, and (**d**) is ref2km. The bold black line represents the boundary of the PR scan range.

### 4.6. Stability of the Algorithm

KNN can classify precipitation types well, but the effect of classifying continuous data is unknown. In fact, continuous data are more widely used and more meaningful. One-month continuous radar data from Lianyungang from 1 July 2012 to 31 July 2012 are used for continuous analysis, and Table 3 shows the result of the continuous analysis.

**Table 3.** Continuous analysis of Lianyungang in July 2012.

| Time(UTC) | Precipitation | POD | FAR | CSI |
|---|---|---|---|---|
| 0:00–12:00 | Stratiform | 0.978 | 0.123 | 0.860 |
|  | Convective | 0.656 | 0.077 | 0.622 |
| 12:00–0:00 | Stratiform | 0.943 | 0.084 | 0.868 |
|  | Convective | 0.784 | 0.153 | 0.687 |

Table 3 shows the result of continuous analysis of Lianyungang in July 2012. The time period of 0:00-12:00(UTC) is daytime in Lianyungang, and the time period of 12:00-0:00(UTC) is evening in Lianyungang. Table 3 shows that both the stratiform and convective classification results are better in evening than in daytime; however, the differences between the daytime and evening results are small, and the classification results are stable in different time periods.

Different geographical conditions may have an impact on the type of precipitation. The precipitation types of Lianyungang, Fuyang and Guangzhou stations were therefore classified to analyze the influence of geographical conditions on KNN. Lianyungang and Fuyang are both located in the subtropical zone, Lianyungang is located near the sea and Fuyang is located inland. The impact of coastal conditions on classification can also be analyzed. Both Guangzhou and Lianyungang are near the sea, Lianyungang is located in the subtropical zone, and Guangzhou is located in the tropical zone; consequently, the influence of latitude conditions on classification can be analyzed. The comparisons of the three sites are shown in Table 4.

**Table 4.** Comparison of the classification results of different geographical conditions.

| Location | Precipitation | POD | FAR | CSI |
|---|---|---|---|---|
| Lianyungang | Stratiform | 0.855 | 0.006 | 0.850 |
|  | Convective | 0.986 | 0.270 | 0.722 |
| Fuyang | Stratiform | 0.869 | 0.012 | 0.859 |
|  | Convective | 0.973 | 0.252 | 0.733 |
| Guangzhou | Stratiform | 0.900 | 0.004 | 0.896 |
|  | Convective | 0.990 | 0.202 | 0.791 |

Table 4 shows the classification results under different geographical conditions. The classification results of Guangzhou have the best performance, and the classification results of Lianyungang have the worst performance. The POD values of the three sites are nearly the same for both precipitation types; Guangzhou has the lowest FAR and highest CSI. However, the CSI values of the three sites show few differences, and the results of classification are stable in different geographical conditions.

*4.7. Overall Analysis*

Table 5 shows the results of the evaluation of the KNN classification results by combining multiple cases of different processes for six Doppler radars in Jiangsu in July, 2012. The POD of KNN for stratiform classification reaches 0.950, the FAR is 0.085, and the CSI is 0.874. From a comprehensive perspective, it is possible to accurately classify more than 85% of the observed stratiform precipitation areas. The POD of the convective classification reaches 0.781, the FAR is 0.137, and the CSI is 0.695. Anagnostou [18] also uses the 2A23 product to classify precipitation types using neural networks, obtaining values of POD = 0.97, FAR = 0.07, and CSI = 0.90 for stratiform precipitation classification and POD = 0.52, FAR = 0.29 and CSI = 0.43 for the classification of convective precipitation. In that paper, the results of SHY95 were also evaluated by 2A23; the stratiform POD, FAR and CSI values were 0.85, 0.05 and 0.81, respectively, and the convective POD, FAR and CSI values were 0.72, 0.59 and 0.36. The cases used are not the same, but with the KNN classification, although

the effectiveness for stratiform precipitation decreased, the classification accuracy of convective precipitation improved significantly.

**Table 5.** Comprehensive evaluation of the KNN classification results.

|  | POD | FAR | CSI |
|---|---|---|---|
| Stratiform | 0.950 | 0.085 | 0.874 |
| Convective | 0.781 | 0.137 | 0.695 |

## 5. Conclusion

A KNN supervised machine learning algorithm is used in this paper to classify precipitation types with ground-based radar data. The ground-based radar data are from Anhui, Jiangsu, Guangdong and Zhejiang Provinces, and the classification results were evaluated using the 2A23 cloud classification product from the TRMM PR at the same time. The KNN algorithm is characterized by high precision, insensitivity to abnormal data, no data input assumptions, and a fast computational speed in the case of small data samples. The method performs well in the classification of precipitation types based on radar data. The radar reflectivity at a height of 2 km and VIL were selected as the classification variables. The values of these two variables in the cases of stratiform precipitation and convective precipitation were compared, and it was found that the two variables differ greatly for the different precipitation types. These two variables and corresponding precipitation types in the 2A23 product were input as training samples in the KNN algorithm. The algorithm calculates the distance between the input samples and the stored training samples (the standardized Euclidean distance was calculated in this paper). The maximum number of classification labels in the k samples closest to the input samples was taken as the classification result for the input samples. Samples can be classified into stratiform precipitation, convective precipitation and other types of precipitation.

Three different precipitation systems (stratiform precipitation, embedded convection, and squall lines) were analyzed. The KNN method is accurate in classifying the location and range of stratiform precipitation and can effectively describe the band arrangement pattern of multiple convective units in squall lines. Moreover, the position and shape of squall lines is well described, and the distribution of convective precipitation and stratiform precipitation is accurately described in the embedded convective systems.

The classification results and accuracy of all cases were analyzed, and the performance of the KNN algorithm in precipitation classification was evaluated. The statistical results confirm the results of the case analysis. Among the overall classification results of many processes and cases, the KNN algorithm is the most accurate in the classification of stratiform precipitation, with a POD of 0.950 and an FAR of only 0.085. The CSI, which reflects the overall classification, reaches 0.874. In all cases, the POD of convective classification is 0.781, the FAR is 0.137, and the CSI is 0.695. The evaluation results indicate that the KNN algorithm can accurately classify almost all stratiform precipitation, and most of the convective precipitation can also be classified accurately.

Because the duration of the radar data is insufficient, it is impossible to study the classification of precipitation types with the KNN algorithm in a certain area over a long period. Although the training and classification cases are limited, the results of the classification in different years and for different regional precipitation types could be important. If long-term radar data from a region were selected, more reliable and accurate classification results could be obtained, and the local climate characteristics and precipitation distribution could be better studied.

**Author Contributions:** Conceptualization, Z.Y. and Y.Y.; methodology, Z.Y., P.L. and Y.Y.; software, Z.Y. and P.L.; investigation, Z.Y. and P.L.; writing—original draft preparation, Z.Y.; writing—review and editing, Z.Y., P.L. and Y.Y.; visualization, Z.Y. and P.L.; supervision, Y.Y.

**Funding:** This research was supported by the National Key Research and Development Program of China (2017YFC1502102) and the National Nature Science Foundation of China (41675098).

**Acknowledgments:** We thank the National Aeronautics and Space Administration (NASA) and Japan Aerospace Exploration (JAXA) for providing the 2A23 TRMM precipitation radar rain characteristics product (https://disc.gsfc.nasa.gov/datasets/TRMM_2A23_V7/summary?keywords=2A23) as the training and evaluating data. We thank the Weather Service Forecast Office of Anhui Province and Jiangsu Province for providing radar data.

**Conflicts of Interest:** The authors declare no conflict of interest.

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
