# Peer review of "Convective/Stratiform Precipitation Classification Using Ground-Based Doppler Radar Data Based on the K-Nearest Neighbor Algorithm"

_remotesensing, doi:10.3390/rs11192277_

Round 1

Reviewer 1 Report

The new version of the manuscript is suitable for pubblication, since te Authors satisfactorily addressed my comments

Author Response

Thank you very much for your time and patience.

Reviewer 2 Report

Review of the revised manuscript “Convective/Stratiform Precipitation Classification Using Ground-based Doppler Radar Data Based on the K-Nearest Neighbor Algorithm” by Yang et al.

The authors have revised their manuscript thoroughly, with additional explanation on both general description and technical details, which have helped the manuscript become more convincing than the previous version. The authors have also improved English language throughout the manuscript, which would be more easily read by the audience of the journal. The issues raised in my previous review have been addressed to a certain extent, although there are still some aspects found remaining to be improved as addressed in the following comments.

Significance and Originality

The authors have added explanations with regards to the importance of studying precipitation type and the advantage of KNN method in general (where these descriptions have certainly improved the manuscript). However, the significance of the present study itself was considered not yet explained sufficiently – Are there any tips in your method or nature of the observation sites that would help explain the originality of this work? How would the algorithm developed contribute to future research (such as in ground-based radar studies)? Are there any significance from the result or outcome of your analysis? Your response to my previous comments on the POD/FAR/CSI scores, arguing that they were superior to those from the previous studies were convincing and I would recommend that those discussion be added in the body of the manuscript.

Adequacy of the Algorithm

Overall, five case studies and one evaluation result seem not enough to show the stability of the present algorithm and therefore to support the adequacy of the algorithm. Analysis such that would show diurnal, seasonal and/or geographical variabilities of the convective/stratiform precipitation classified are recommended, not only to show the algorithm stability but also to improve the scientific quality of the present study.

Selecting Ref2km

The selection of the geophysical parameters is essential in a machine learning-based method, affecting its performance directly. In this study, the authors selected radar reflectivity at 2km height (ref2km) as one of the two parameters to classify the precipitation type, explaining:

In some cases, the temperature at 2 km in the vertical height layer is close to 0 °C, and there is a mixture of liquid and solid phase water and transitions between the two phases.

However, the height of the melting layers and bright bands vary over the globe as shown in previous works such as Gao et al. (2017). Additional description is recommended, explaining the reason for selecting ref2km to classify the precipitation types at the five specific observation sites.

Gao, J.; Tang, G.; Hong, Y. Similarities and Improvements of GPM Dual-Frequency Precipitation Radar (DPR) upon TRMM Precipitation Radar (PR) in Global Precipitation Rate Estimation, Type Classification and Vertical Profiling. Remote Sens. 2017, 9, 1142.

TRMM Precipitation Type

The convective/stratiform classification from the TRMM 2A23 product seems somewhat unrealistically scattered, especially in Figures 4(a) and 5(a). Would they affect the KNN method construction when you train it using these data? If so, have you considered a way to eliminate those unrealistic data, avoiding them to be included in the training dataset?

Author Response

Response to Reviewer

The authors have revised their manuscript thoroughly, with additional explanation on both general description and technical details, which have helped the manuscript become more convincing than the previous version. The authors have also improved English language throughout the manuscript, which would be more easily read by the audience of the journal. The issues raised in my previous review have been addressed to a certain extent, although there are still some aspects found remaining to be improved as addressed in the following comments.

Response: Thanks a lot for your time and encouraging comments. We made substantial revision to improve our manuscript following the reviewer’s suggestions, and commissioned AJE company to retouch the language of the manuscript. Please see the responses to each comment below. We believe all the reviewer’s comments are appropriately addressed.

Point 1. Significance and Originality

The authors have added explanations with regards to the importance of studying precipitation type and the advantage of KNN method in general (where these descriptions have certainly improved the manuscript). However, the significance of the present study itself was considered not yet explained sufficiently – Are there any tips in your method or nature of the observation sites that would help explain the originality of this work? How would the algorithm developed contribute to future research (such as in ground-based radar studies)? Are there any significance from the result or outcome of your analysis? Your response to my previous comments on the POD/FAR/CSI scores, arguing that they were superior to those from the previous studies were convincing and I would recommend that those discussions be added in the body of the manuscript.

Response: Thank you for your comment. In this study, TRMM 2A23 products are used as training samples, and KNN is used to classify the precipitation types of ground-based radar. Machine learning is a kind of new algorithm, and it is seldom used to classify precipitation types. KNN is a mature classification algorithm with many advantages in many fields, but there is no relevant study to prove the applicability of KNN in the classification of precipitation types, and this study tries to use KNN to classify precipitation types and explore the applicability of KNN in the classification of precipitation types. this part has been added in line 137-141 in manuscript.

The results of the classification are not bad and the classification speed is fast, which indicates that the machine learning algorithm has great potential in the classification of precipitation types. if more relevant variables and more suitable algorithms are selected, the accuracy of precipitation classification can be further improved. And accurate classification of precipitation types is helpful to understand local meteorological characteristics. Because the duration of the radar data is insufficient, it is impossible to study the classification of precipitation types with the KNN algorithm in a certain area over a long period. Although the training and classification cases are limited, the results of the classification in different years and for different regional precipitation types could be important. If long-term radar data from a region were selected, more reliable and accurate classification results could be obtained, and the local climate characteristics and precipitation distribution could be better studied.

And the response to previous comment on the POD/FAR/CSI scores has been added in line 441-448 in the manuscript.

Point 2. Adequacy of the Algorithm

Overall, five case studies and one evaluation result seem not enough to show the stability of the present algorithm and therefore to support the adequacy of the algorithm. Analysis such that would show diurnal, seasonal and/or geographical variabilities of the convective/stratiform precipitation classified are recommended, not only to show the algorithm stability but also to improve the scientific quality of the present study.

Response: Yes, only five different types of precipitation are analyzed in this manuscript, and the amount of data is a little small. However, the radar stations of these cases are different, and KNN can classify precipitation types with good effect. The data from May to September in Nanjing is selected for the overall classification evaluation. The radar site has not changed, but it contains different seasons. These seasons have more convective weather, the rare conditions of convective weather in winter can be avoided. At the same time, these data include diurnal changes, which indicates that KNN has better stability and reliability in different seasons and time. If a large amount of data is used, the stability and versatility of KNN will be better demonstrated.

Point 3. Selecting Ref2km

The selection of the geophysical parameters is essential in a machine learning-based method, affecting its performance directly. In this study, the authors selected radar reflectivity at 2km height (ref2km) as one of the two parameters to classify the precipitation type, explaining:

In some cases, the temperature at 2 km in the vertical height layer is close to 0 °C, and there is a mixture of liquid and solid phase water and transitions between the two phases.

However, the height of the melting layers and bright bands vary over the globe as shown in previous works such as Gao et al. (2017). Additional description is recommended, explaining the reason for selecting ref2km to classify the precipitation types at the five specific observation sites.

Gao, J.; Tang, G.; Hong, Y. Similarities and Improvements of GPM Dual-Frequency Precipitation Radar (DPR) upon TRMM Precipitation Radar (PR) in Global Precipitation Rate Estimation, Type Classification and Vertical Profiling. Remote Sens. 2017, 9, 1142.

Response: Thank you for your comment. Additional description of the variation of melting layers and bright band height over globe has been added in line 224-228 of the manuscript and the paper (Gao, Tang, & Hong, 2017) has been referenced: The reflectivity of 2km height can distinguish most of the convective precipitation and stratiform precipitation (Anagnostou, 2004). But if the bright band is present, the reflectivity will increase significantly, which will have a bad influence on the classification results. The bright band is not expected to appear at the time of classification. 2-km altitude is high enough to provide amount of radar data out to about 150 km radius and 2-km altitude is low enough not to be seriously affected by the bright band, which is usually appears at the height of 2.5 km to 4.5 km in tropical and sub-tropical area (Gao et al., 2017).

Point 4. TRMM Precipitation Type

The convective/stratiform classification from the TRMM 2A23 product seems somewhat unrealistically scattered, especially in Figures 4(a) and 5(a). Would they affect the KNN method construction when you train it using these data? If so, have you considered a way to eliminate those unrealistic data, avoiding them to be included in the training dataset?

Response:

Yes, some data in 2A23 products have obvious errors, there are still steps needed to make the comparisons of two datasets meaningful. These data should be eliminated: (1) a pixel is classified as stratiform by the 2A23 product if a BB is not detected and ref2km is greater than 40 dBz or if there is a BB detected and ref2km is greater than 42 dBz with a horizontal gradient greater than 3 dB/km; (2) a pixel is classified as convective by the 2A23 product if no BB is detected but ref2km is less than 40 dBz; and (3) a pixel is classified as convective by the 2A23 product if a BB is detected. This part shows in the manuscript in line 187-192.

Reference:

Anagnostou, E. N. (2004). A convective/stratiform precipitation classification algorithm for volume scanning weather radar observations. Meteorological Applications, 11(4), 291-300.

Gao, J., Tang, G., & Hong, Y. (2017). Similarities and improvements of GPM Dual-Frequency Precipitation Radar (DPR) upon TRMM precipitation radar (PR) in global precipitation rate estimation, type classification and vertical profiling. Remote Sensing, 9(11), 1142.

Reviewer 3 Report

I am happy that the authors adressed all my previously raised issues and the manuscript is now ready for publication once few minor changes are done. The few comments can be found in the attached file. I would also suggest to check the definition of all the Figures, as some of them look blur (at least on my pdf)

Author Response

Response:Thanks a lot for your time and encouraging comments. The language error marked in the manuscript has been corrected, and the formula has adopted a larger font size.

Round 2

Reviewer 2 Report

Review of the second revision of “Convective/Stratiform Precipitation Classification Using Ground-based Doppler Radar Data Based on the K-Nearest Neighbor Algorithm” by Yang et al.

Since the last revision, the manuscript has improved to some extent, especially in the introduction part; however, the algorithm design in this manuscript is found still questionable and the results shown are very limited, where the scientific quality is considered not sufficient to be published for Remote Sensing. Therefore, I would recommend to keep it as a major revision.

As commented in my last review, the results presented here are based on only five case studies and one evaluation, which was considered not enough to show the adequacy of the algorithm and elaborate scientific content/discussions that would be of interest to the readers of the journal. Consequently, I would strongly recommend additional analysis - such as algorithm sensitivity tests or retrieved precipitation type analysis - to be included in the manuscript. Certainly, the limitation of the observation period would make this difficult; however, given that some sites were operated for a month, you may discuss, for example, on the diurnal variabilities or geographical differences among the sites.

The authors selected the radar reflectivity at 2 km height (ref2km) as one of the two predictors in the algorithm based on a work by Anagnostou (2004) who noted that this parameter “seems to best distinguish the different precipitation types”. Please discuss (with other references, if necessary) why this parameter can be applied to the precipitation systems in China, given that Anagnostou (2004) investigated for those in the United States. In fact, Anagnostou (2004) used six predictors to classify the precipitation types. Please describe the reason for selecting the two parameters in this work and the adequacy of your selection.

A few specific comments;

1. What are the number of scenes used for training and evaluation? Please add the numbers in Table 1.

2. “The ground-based radar data are from Anhui and Jiangsu Provinces from 2007 to 2012” – this sentence in the conclusion is misleading as it could give an impression that the observations were conducted continuously for 5 years, whereas the actual period seems to be around 3 months in total of the five sites (according to Table 1). Please rephrase the sentence.

Anagnostou, E. N. (2004). A convective/stratiform precipitation classification algorithm for volume scanning weather radar observations. Meteorological Applications, 11(4), 291-300.

Author Response

Response to Reviewer

Since the last revision, the manuscript has improved to some extent, especially in the introduction part; however, the algorithm design in this manuscript is found still questionable and the results shown are very limited, where the scientific quality is considered not sufficient to be published for Remote Sensing. Therefore, I would recommend to keep it as a major revision.

Response: Thanks a lot for your time and comments. We made substantial revision to improve our manuscript following the reviewer’s suggestions. Please see the responses to each comment below. We believe all the reviewer’s comments are appropriately addressed.

Point 1: As commented in my last review, the results presented here are based on only five case studies and one evaluation, which was considered not enough to show the adequacy of the algorithm and elaborate scientific content/discussions that would be of interest to the readers of the journal. Consequently, I would strongly recommend additional analysis - such as algorithm sensitivity tests or retrieved precipitation type analysis - to be included in the manuscript. Certainly, the limitation of the observation period would make this difficult; however, given that some sites were operated for a month, you may discuss, for example, on the diurnal variabilities or geographical differences among the sites.

Response: Thank you for your comment. Continuous analyze has been added in the manuscript in line 412-437 as follow:

KNN can classify precipitation types well, but the effect of classifying continuous data is unknown. In fact, continuous data are more widely used and more meaningful. One-month continuous radar data from Lianyungang from 2012.7.1 to 2012.7.31 are used for continuous analysis, Table 3 shows the result of the continuous analysis.

Table 3 continuous analysis of Lianyungang in 2012.7

Time(UTC)

Precipitation

POD

FAR

CSI

0:00-12:00

Stratiform

0.978

0.123

0.860

Convective

0.656

0.077

0.622

12:00-0:00

Stratiform

0.943

0.084

0.868

Convective

0.784

0.153

0.687

Table 3 shows the result of continuous analysis of Lianyungang in July, 2012. The time period of 0:00-12:00(UTC) is daytime in Lianyungang, and the time period of 12:00-0:00(UTC) is evening in Lianyungang. Table 3 shows that classification results of both stratiform and convective are better in evening than in daytime, however, the results of daytime and evening have little differences, the results of classification are stable in different time periods.

Different geographical conditions may have an impact on the type of precipitation, the precipitation types of Lianyungang, Fuyang and Guangzhou stations were classified to analyze the influence of geographical conditions on KNN. Lianyungang and Fuyang are both located in the subtropical zone, Lianyungang is located near the sea and Fuyang is located inland, the impact of coastal conditions on classification can be analyzed, both Guangzhou and Lianyungang are near the sea, Lianyungang is located in the subtropical zone and Guangzhou is located in the tropical zone, so that the influence of latitude conditions on classification can be analyzed. The comparison of three sites are shows in Table 4.

Table 4 comparison of the classification results of different geographical conditions

Location

Precipitation

POD

FAR

CSI

Lianyungang

Stratiform

0.855

0.006

0.850

Convective

0.986

0.270

0.722

Fuyang

Stratiform

0.869

0.012

0.859

Convective

0.973

0.252

0.733

Guangzhou

Stratiform

0.900

0.004

0.896

Convective

0.990

0.202

0.791

Table 4 shows the classification results under different geographical conditions. The classification results of Guangzhou have the best performance, and the classification results of Lianyungang have worst performance. The POD of three sites are nearly the same in both precipitation types, Guangzhou has the lowest FAR, and the highest CSI. However, the CSI of three sites have little differences, the results of classification are stable in different geographical conditions.

Point 2: The authors selected the radar reflectivity at 2 km height (ref2km) as one of the two predictors in the algorithm based on a work by Anagnostou (2004) who noted that this parameter “seems to best distinguish the different precipitation types”. Please discuss (with other references, if necessary) why this parameter can be applied to the precipitation systems in China, given that Anagnostou (2004) investigated for those in the United States. In fact, Anagnostou (2004) used six predictors to classify the precipitation types. Please describe the reason for selecting the two parameters in this work and the adequacy of your selection.

Response: Thank you for your comment. (Yang, Chen, & Qi, 2013) used radar data of Heifei, China for classification of precipitation types. They used ref2km, VIL, the product of radar top height and reflectivity value at 2km (units in km · dBz), and standard deviation of ref2km for classification. These features are found to best discriminate between convective and stratiform precipitation (Yang et al., 2013). In their four features, the standard deviation of ref2km and the product of radar top height and reflectivity value at 2km are based on the ref2km. The product of radar top is interpolated from the whole layers of reflectivity, and always have missing values in the area near the radar station, which will have a negative impact on the classification results in this area. Thus, only ref2km and VIL are chosen for the classification in this research, ref2km could provide amount of radar data out to about 150 km radius and it is low enough not to be seriously affected by the bright band, the VIL could reflect the change of the convective system and will not increase significantly when bright band is detected, which could reduce the influence of bright band.

Point3. What are the number of scenes used for training and evaluation? Please add the numbers in Table 1.

Response: Thank you for your comment, the number of scenes used for training and evaluation has been added in Table 1 in line 193 as follow:

Table 1 Radar site information and data usage time

Station

Date

Coordinate

Usage

Case number

Hefei

2010.06.06-2010.06.10

117.258°E, 31.867°N

Classification

2

Fuyang

2005.06.25-2005.06.26

2007.07.07-2007.07.09

115.741°E, 32.879°N

Training and Classification

4

Lianyungang

2012.07.01-2012.07.31

119.294°E, 34.651°N

Training and Classification

7

Nanjing

2012.07.01-2012.07.31

118.698°E, 32.191°N

Classification

5

Guangzhou

2008.06.04-2008.06.13

120.976°E, 32.076°N

Training and Classification

4

Wenzhou

2008.06.04-2008.06.13

117.152°E, 34.293°N

Classification

2

Point 4. “The ground-based radar data are from Anhui and Jiangsu Provinces from 2007 to 2012” – this sentence in the conclusion is misleading as it could give an impression that the observations were conducted continuously for 5 years, whereas the actual period seems to be around 3 months in total of the five sites (according to Table 1). Please rephrase the sentence.

Response: Thank you for your comment, the sentence has been rephrased in line 456-457 in the manuscript as follow:

The ground-based radar data are from Anhui, Jiangsu, Guangdong and Zhejiang Provinces.

Reference:

Yang, Y., Chen, X., & Qi, Y. (2013). Classification of convective/stratiform echoes in radar reflectivity observations using a fuzzy logic algorithm. Journal of Geophysical Research Atmospheres, 118(4), 1896–1905.

This manuscript is a resubmission of an earlier submission. The following is a list of the peer review reports and author responses from that submission.

Round 1

Reviewer 1 Report

the paper "Convective/Stratiform Precipitation Classification Using Ground-based Doppler Radar Data Based on the K-Nearest Neighbor Algorithm" by Yang and co-workers, presents a statistical technique to classify precipitation as convective or stratiform, by using weather radar data.

The paper is interesting, even if this topic is discussed in dozens of paper, well written, and the results seem rather good. I think however, that a number of modifications should be made on the manuscript before the final submission. I list below some suggestion to improve the paper.

Introduction.

Among the methods used to discriminate C/S precipitation should be also cited the use of standard deviation Bringi, et al, 2003 J. Atmos. Sci. 60, 354–365).

lines 112-139. This long introduction on machine learning approaches should be much shortened, especially the historical perspective, out of the scope of the present work.

Data description.

The acquisition geometry of ground based radars and PR are quite different, resulting in cells of different shapes and volume. The "numerical interpolation" mentioned at line 166 should be much better described, taking also into account that the size of the radar cells changes with the distance from the radar. Following the description on lines 195-197, it seems these differences are not taken into account. Moreover, the PR is a Ku band radar, while the CINRAD radars are S-band, with different sensitivity and attenuation properties: does this make any intrinsic difference? Finally, the Authors uses to train and validate the algorithm the product 2A23, which is a satellite product, quite far from a reference truth. The algorithm is designed to perform close to the TRMM-PR product, but the Authors should convince the reader that the TRMM product is good enough to be considered a reference. Usually, satellite products are validated by ground instrument (e.g. disdrometers), the opposite is more rarely found. 

Results.

lines 252-255. This sentence is not clear, please, rewrite it more clearly.

in table also the scores for "others" class should be reported: probably it refers to cells where the precipitation is mixed, that can be expected, given the scale of variability of precipitation structure (See Tokay et al, 2017, Journal fo hydrometeorology, 18, 3165-3179).  

Reviewer 2 Report

Review of the article “Convective/Stratiform Precipitation Classification Using Ground-based Doppler Radar Data Based on the K-Nearest Neighbor Algorithm” by Yang et al.

The manuscript developed a method to distinguish convective and stratiform precipitation from ground-based radar measurements. The classification method was based on the k-nearest neighbor algorithm, trained by the 2A23 product of the Tropical Precipitation Measurement Mission (TRMM) satellite.

Given the previous attempts of precipitation type classification documented in the introduction, the manuscript had very little explanation on the significance and originality of the present study. In their methodology, authors claimed that the two variables, radar reflectivity at 2 km height (denoted as ref2km) and vertical integrated liquid water (denoted as VIL), showed different characteristics among the convective and stratiform types, and therefore, can be used to distinguish the two precipitation types; however, I find them rather identical, especially in the reflectivity range between 20-40 dbz and VIL below 3 kg/m2 in Figure 1. Depending on the nature of precipitation processes, ref2km and VIL are highly likely to share common values, which would be inappropriate to be incorporated to classify the convective/stratiform types. Parameters such as detection of bright band (in an explicit manner, rather than in an implicit form of ref2km as described in this manuscript) would adequately explain, in a more physical context, to distinguish the two precipitation types. Further, the study used the same 2A23 product for training and evaluating, which is quite questionable, and independent observation data are desired for evaluation to illustrate the adequacy of the classification results. Unfortunately, the probability of classification of 68% for convective type is not convincing result despite its detection being the core of this work. Therefore, it is recommended that the manuscript be rejected at this time.